# Prilling as an Effective Tool for Manufacturing Submicrometric and Nanometric PLGA Particles for Controlled Drug Delivery to Wounds: Stability and Curcumin Release

**DOI:** 10.3390/pharmaceutics17010129

**Published:** 2025-01-17

**Authors:** Chiara De Soricellis, Chiara Amante, Paola Russo, Rita Patrizia Aquino, Pasquale Del Gaudio

**Affiliations:** Department of Pharmacy, University of Salerno, Via Giovanni Paolo II 132, 84084 Fisciano, Italy

**Keywords:** prilling, solvent evaporation, nanoparticles, submicrometric particles, PLGA, curcumin

## Abstract

Background/Objectives: This study investigates for the first time the use of the prilling technique in combination with solvent evaporation to produce nano- and submicrometric PLGA particles to deliver properly an active pharmaceutical ingredient. Curcumin (CCM), a hydrophobic compound classified under BCS (Biopharmaceutics Classification System) class IV, was selected as the model drug. Methods: Key process parameters, including polymer concentration, solvent type, nozzle size, and surfactant levels, were optimized to obtain stable particles with a narrow size distribution determined by DLS analysis. Results: Particles mean diameter (d_50_) 316 and 452 nm, depending on drug-loaded cargo as Curcumin-loaded PLGA nanoparticles demonstrated high encapsulation efficiency, assessed via HPLC analysis, stability, and controlled release profiles. In vitro studies revealed a faster release for lower drug loadings (90% release in 6 h) compared to sustained release over 7 days for higher-loaded nanoparticles, attributed to polymer degradation and drug-polymer interactions on the surface of the particles, as confirmed by FTIR analyses. Conclusions: These findings underline the potential of this scalable technique for biomedical applications, offering a versatile platform for designing drug delivery systems with tailored release characteristics.

## 1. Introduction

Nano- and submicrometric particles, defined as particles with diameters smaller than 1 μm (10–1000 nm), can effectively deliver both hydrophobic and hydrophilic drugs to various regions of the body in a controlled way over time [1]. In fact, as drug delivery systems, polymeric nano- and submicrometric particles have achieved significant success due to their properties conferred by their small size [2]. The drug can be dissolved, entrapped, encapsulated, or attached to a nanoparticle matrix. They have a large surface area capable of binding different drugs; their smaller size, high tissue diffusivity, and ability to overcome biological barriers facilitate tissue penetration and accumulation at the target site [3,4,5]. These delivery systems show several advantages compared to larger particles, including high and easy drug encapsulation [6], improved drug safety and efficiency, reduced toxic side effects, and enhanced patient compliance [7,8,9,10]. Moreover, they can increase the solubility of poorly soluble drugs, enhance pharmacokinetic profiles and bioavailability, and maintain consistent drug concentration over time [11,12,13].

Different methods can be used to produce nanoparticles, including solvent emulsification, solvent diffusion [14], salting out [15,16], nanoprecipitation [17], and microfluidics [18,19].

Prilling, or laminar jet breakup, is an encapsulation technique used to produce beads and microparticles [20,21]. It is based on the breaking apart of a laminar liquid jet into a series of uniform droplets by applying a mechanical vibration to it [22]. When the vibration wavelength reaches its critical value, namely the Rayleigh wavelength, the laminar jet breaks apart into a chain of uniform droplets [23]. The droplets are mainly transformed into beads or microparticles upon contact with the collecting solution. Prilling offers several advantages compared to other encapsulation techniques, enabling the production of microparticles or beads with narrow size distribution and spherical shape as well as high encapsulation efficiency in a single-step process [24,25,26]. The shape and size distribution depend on solution properties (density, dynamic viscosity, and the nature of the collecting solution), the flow rate of the feed solution, nozzle diameter, and the frequency of vibration [27,28].

Dynamic viscosity, or nozzle viscosity, influences the laminar flow breakup process, as higher viscosity solutions can resist the vibration needed to break the jet into droplets. The flow rate must be carefully controlled to prevent spray phenomena and droplet deformation upon entering the collecting solution, which can lead to coalescence [29]. Droplet deformation is influenced by the surface tension of the collecting solution; to minimize this effect, the distance between the nozzle and the impact size can be reduced, or a surfactant can be added [30]. Higher vibration frequencies during the prilling process can lead to the generation of satellite droplets, resulting in a broader size distribution. To avoid this phenomenon, the frequency is kept as low as possible [31].

Other than the production method, polymeric excipients play the most important role in defining the final properties of the produced particles.

Poly (lactic-co-glycolic) acids, or PLGA, are copolymers of polylactic acid (PLA) and polyglycolic acid (PGA), approved by the United States Food and Drug Administration (FDA) and widely used among biodegradable and biocompatible polymers as drug delivery systems, tissue engineering, and healing of bone defects [32,33]. PLGA enables the controlled and sustained release of active pharmaceutical ingredients, prolonged residence time, targeted delivery, and exhibits low toxicity [34]. Additionally, PLGA promotes wound healing as the monomeric lactic acid component contributes to accelerating the angiogenesis and the wound repair [35]. Several studies have shown that various techniques can be used to produce PLGA nanoparticles loaded with both hydrophobic and hydrophilic drugs in order to obtain a controlled and sustained release [36,37,38].

The primary aim of this work was to assess the feasibility of using the prilling technique to produce, for the first time, submicrometric and nanometric PLGA particles loaded with bioactive molecules for controlled release. This approach involved combining prilling with solvent evaporation to obtain stable formulations using biodegradable and biocompatible hydrophobic polymers. A secondary objective has been to demonstrate the capability of the produced particles to deliver properly an active pharmaceutical ingredient. In order to do so, curcumin (CCM), a hydrophobic compound classified under BCS (Biopharmaceutics Classification System) class IV, was selected as the model drug. Extracted from the rhizome of *Curcuma longa*, curcumin exhibits a wide range of biological activities, including antibacterial, antifungal, antiviral, antioxidant, anti-inflammatory, and anticancer properties. Additionally, curcumin plays a crucial role in wound healing by modulating the inflammatory and proliferative phases and promoting angiogenesis [39,40]. However, curcumin has notable drawbacks, such as low solubility, photosensitivity, instability, and poor bioavailability [41,42]. These limitations can potentially be mitigated through proper encapsulation and controlled release from delivery systems like PLGA nanoparticles [43].

## 2. Materials and Methods

Poly (D, L-lactide-co-glycolide) ester terminated (Mw 50,000–75,000, 85:15) was purchased from Sigma Aldrich (Milan, Italy). Polyoxyethylenesorbitan Trioleate, Tween 85, oleic acid, ≥50% (balance primarily C10-C18 fatty acids) was obtained from Sigma Aldrich (Milan, Italy). Poly (vinyl alcohol) (Mw 9000–10,000, 80% hydrolyzed, and Mw 31,000–50,000, 98–99% hydrolyzed) was purchased from Sigma Aldrich (Milan, Italy). Ethyl acetate and dichloromethane (VWR Chemicals, Briare, France) were used for the dissolution medium. Curcumin was donated from the laboratory of Professor Manuela Rodriquez, Department of Pharmacy, Naples [44]. Acetonitrile for HPLC (VWR Chemicals, Briare, France), water for HPLC (Carlo Erba Reagents, Val de Reuil Cedex, France), formic acid (Sigma Aldrich, Milan, Italy), and methanol for HPLC (Carlo Erba Reagents, Val de Reuil Cedex, France) were used as the mobile phase for high-performance liquid chromatography (HPLC). Polyoxyethylenesorbitan monooleate, Tween 80 viscous liquid, from Sigma Aldrich (Milan, Italy).

Potassium chloride (Sigma Aldrich, Milan, Italy), sodium phosphate dibasic Reagent Plus ≥ 99.0% (Sigma Aldrich, Milan, Italy), sodium phosphate monobasic Reagent Plus ≥ 99.0% (Sigma Aldrich, Milan, Italy), and sodium chloride (Sigma Aldrich, Milan, Italy) were used for PBS 7.44. Spectra/Por 6 Dialysis Membranes, Pre-wetted RC Tubing, 3.5 kDa MWCO (Thermo Fisher Scientific Inc., Waltham, MA, USA) were used for the release studies of curcumin.

### 2.1. Nano- and Submicrometric Particles Preparation

Particles were produced through the prilling technique using the Büchi Encapsulator B-390 (Büchi Labortechnik AG, St. Gallen, Switzerland). PLGA, at different concentrations (1.0; 2.5; and 5.0% *w*/*v*), was dissolved either in ethyl acetate (EtOAC) or dichloromethane (DCM). Tween 85 at different concentrations (0.5; 1.5; 2.5; 3; and 3.5% *v*/*v*) was added to the solution to avoid the aggregation of particles. A laminar jet of polymer solution is broken into droplets by vibration, falling down into the collecting solution composed of an aqueous solution of polyvinyl alcohol (PVA) in different concentrations (0.5; 1.0; and 2.0% *w*/*v*). Experiments were performed using different nozzles (80; 120; and 200 μM) as shown in Table 1.

The distance between the nozzle and the collecting solution was fixed at 8 cm. The particles formed were held in the collecting solution under gentle stirring for 48 h to promote solvent evaporation.

### 2.2. Morphology and Particle Size Distribution

Morphological analysis was conducted by scanning electron microscopy (SEM) (Tescan Orsay Holding, Brno, Czech Republic). Some drops of nanosuspension were dispersed on a carbon-coated aluminum stub (Agar Scientific, Stansted, UK) and placed under nitrogen. The samples were analyzed without undergoing metallization.

Particle size distribution and mean diameter were evaluated by static light scattering (SLS) (N5, Beckman Coulter, Miami, FL, USA), taking the average of three measurements for each sample. Results were expressed as mean diameter (d_50_) ± standard deviation (SD) of at least three independent experiments.

### 2.3. Preparation of Feed Solution Containing Curcumin

Two different concentrations of curcumin (10–20% *w*/*w*) were dissolved in EtOAc and added to PLGA 1% (*w*/*v*) and Tween 85 1.5% (*v*/*v*). PLGA particles loaded with CCM were prepared using the prilling technique (Encapsulator Büchi 390). The drops falling into the collecting solution composed of an aqueous solution of PVA_LMW_ 0.5% (*w*/*v*) solidify by solvent evaporation to form PLGA nanoparticles. The parameters used are as follows:-Frequency: 270 Hz;-Flow rate: 8 mL/min;-Nozzle: 200 μM.

The nanosuspension was centrifuged for 20 min at 30,000 rpm; the supernatant was washed twice in distilled water by centrifugation in the same conditions. The nanoparticles were then freeze-dried for further characterization (LyovaporTM L-200, Buchi, Italy).

### 2.4. Nanoparticles Characterization

#### 2.4.1. Dynamic Laser Scattering

The size distribution and surface potential of blank and loaded PLGA nanoparticles were determined using the Litesizer^TM^ 500 Particle Analyzer (Anton Paar, Graz, Austria). Data acquisition was performed through the Kalliope^TM^ software v.2.34.3.

Particle size and polydispersity index (PDI) were measured by diluting all samples (1:100) with milli-Q water to ensure the measure of size distribution. The ζ–potential was measured by diluting the samples in the same way in milli-Q water. The Litesizer^TM^ 500 used the Smoluchowski approximation to determine zeta potential from electrophoretic mobility measurements based on Henry’s equation. At different time points (0 and 28 days), particle size, PDI, and ζ–potential were measured to study the stability of all PLGA nanoparticles in a colloidal suspension storage at 4 °C.

#### 2.4.2. FT-IR Analysis

The possible interactions between the polymer and the drug in the different formulations were studied through FT-IR analyses, performed using an FT-IR spectrophotometer (Spotlight 400N FT-NIR Imaging System, Perkin Elmer Inc., Waltham, MA, USA) equipped with an ATR accessory (ZnSe crystal plate). The spectra were obtained at room temperature in the range 4000–600 cm^−1^ with a resolution step of 1.0 cm^−1^ and 128 scans.

#### 2.4.3. Drug Content and Encapsulation Efficiency

HPLC (Agilent 1100 Series HPLC, Agilent Technologies, Santa Clara, CA, USA) with a UV detector set at 428 nm was used to evaluate the amount of curcumin encapsulated into PLGA nanoparticles. The separation was performed using a reversed-phase column (NUCLEODUR 100-5 C18 EC 250 mm; 4 mm; 5 μM) and a mixture of water with formic acid (B) and methanol (B) as the mobile phase. The volume ratio of solvent A against solvent B is 20:80, the flow rate is 1 mL/min, and the injection volume is 20 μL. About 13 mg of PLGA nanoparticles loaded with curcumin was dissolved in 3 mL of CH_3_CN/CH_3_OH 70:30, sonicated for 10 min, and filtered with 0.45 μM. The results obtained were compared with a calibration curve in the range of 1 μg/mL and 500 μg/mL (R^2^ 0.9991).

#### 2.4.4. In Vitro Drug Release

The release of curcumin from PLGA nanoparticles was evaluated using a dialysis membrane (3.5 kDa molecular weight cut-off). A proper amount of lyophilized PLGA nanoparticles loaded with curcumin was dissolved into PBS pH 7.4. The dialysis membrane was suspended into a mixture of PBS and MeOH (70:30) and 0.1% of Tween 80. Curcumin alone was dissolved in MeOH. Both systems were kept at 37 °C under stirring (110 rpm) using an orbital shaker (SKI 4, Argolab, Carpi, Italy) for 24 h. At specific time points, 1 mL of the sample was taken and replaced by an equal volume of fresh medium. The released curcumin was analyzed by HPLC, following the method described above.

## 3. Results and Discussion

### 3.1. PLGA Nanoparticles Preparation and Characterization

PLGA nanoparticles were produced by the prilling technique, setting proper operational parameters to break up the jet of polymer solution coming out of the nozzle (Encapsulator Buchi 390). Several parameters were investigated to obtain PLGA nanoparticles in a narrow size distribution and in spherical shape, such as the organic solvent used for the dissolution of PLGA, the PVA and TWEEN 85 concentration in the collecting solution, as well as the nozzle size and the PLGA concentration for the prilled solution.

After a first series of preliminary experiments used to set some of the operational parameters, the impact of the organic solvent on PLGA mean particle size was investigated. DCM (immiscible in water), one of the most commonly used solvents to prepare PLGA nanoparticles by the solvent evaporation method [45], and EtOAc (slightly soluble in water) were used. Figure 1 shows the mean particle size distributions of PLGA nanoparticles prepared by using optimized conditions with the two different solvents, using 1.0% (*w*/*v*) PLGA solution and 3.0% (*v*/*v*) Tween 85 using a 200 µM nozzle, while PVA at different molecular weights, namely PVA_MMW_ and PVA_LMW_, was used as 1.0% (*w*/*v*) collecting solutions.

The use of EtOAc as an organic solvent resulted in a narrower particle size distribution compared to DCM. This effect can be attributed to differences in interfacial tension between the polymeric feed solution and the PVA collecting solution, which is lower when EtOAc is used [4]. Scanning Electron Microscopy (SEM) analysis revealed differences in the tendency of nanoparticles to aggregate. In fact, as shown in Figure 2a, nanoparticles obtained by using DCM as a feed solvent have very small size (<50 nm). In contrast, particles produced using EtOAc as a feed solvent exhibited reduced aggregation tendencies, forming well-defined particle suspensions that remained stable over time. To avoid aggregation issues, the study was conducted using only EtOAc as the solvent for the PLGA feed solutions.

PVA is the most commonly used emulsifier in the formation of PLGA nanoparticles [45]; therefore, the influence of PVA concentration in the collecting bath on PLGA particle properties was also investigated. Two different concentrations, 0.5% to 1.0% (*w*/*v*) and two molecular weights (namely PVA_LMw_ and PVA_MMw_) were used to prepare the collecting solution. As shown in Figure 1B, an increase in PVA concentration from 0.5 to 1.0% led to an increase in both particle size and size distribution of the PLGA nanoparticles. This effect is probably due to the interpenetration of PVA and PLGA molecules within the particle structure during the evaporation of organic solvent [46]. The use of PVA_MMW_ enhances the interaction between PVA and PLGA chains, resulting in a higher presence of residual surfactants. Numerous studies have shown that the use of PVA_MMW_ tends to produce unstable suspensions [46,47,48]; thus, to improve the stability of the nanosuspensions, PVA_LMW_ was preferred for further investigations.

TWEEN 85 is a non-ionic surfactant, able to interact with hydrophobic and hydrophilic components, commonly used to stabilize nanosuspensions, preventing the aggregation by forming a steric barrier around the nanoparticles [49]. The effect of TWEEN 85 concentration on the size of PLGA nanoparticles was also investigated, using concentrations ranging from 0.5% to 3.5% (*v*/*v*). Although variations in surfactant concentration did not significantly affect particle size or size distribution, a lower concentration of 0.5% (*v*/*v*) was associated with a reduced yield of nanoparticles. To optimize both stability and excipient efficiency, a final TWEEN 85 concentration of 1.5% (*v*/*v*) was selected. Subsequently, the influence of the nozzle on the particle size of PLGA nanoparticles was evaluated. Three nozzles were tested, 80 µM, 120 µM, and 200 µM, under the optimized operational parameters shown in Table 1. As expected, reducing the nozzle diameter led to the formation of smaller particles, which exhibited a tendency to aggregate in the aqueous medium (Appendix A).

Following the optimization of other prilling process variables that affect nanoparticle properties, the concentration of PLGA in the feed was also investigated. Using a 200 µm nozzle, PVA_LMw_ 0.5% (*w*/*v*) and TWEEN 85 1.5% (*w*/*v*) were used to avoid particle aggregation, and PLGA at various concentrations, ranging from 1.0 to 5.0% (*w*/*v*), was used to produce nanoparticles. The nanoparticle size increased slightly with higher concentrations of PLGA. This increase in size may be attributed to the larger droplet size formed at the nozzle tip, caused by the increased viscosity of the organic phase due to the higher polymer content during the breakup of the laminar jet in the prilling process [47].

As PLGA concentration rose, coalescence and particle agglomeration phenomena were observed. This is likely due to the insufficient amount of PVA in the collecting solution being unable to adequately separate PLGA droplets upon contact, resulting in these undesirable effects (Appendix A) [48]. The d_50_ and SD (±) values of the colloidal suspensions obtained using different operating parameters were reported in Appendix A.

### 3.2. PLGA Nanoparticles Loaded with Curcumin

Several studies have reported the use of PLGA nanoparticles to encapsulate curcumin for the treatment of different diseases, cancer, and wound healing [50,51].

To investigate the impact of different curcumin concentrations on the physicochemical properties of the nanoparticles, two specific curcumin amounts were tested. Curcumin (CCM) at 10% and 20% *w*/*w* was incorporated into PLGA nanoparticles by directly adding CCM to the PLGA feed solution, followed by prilling under optimized operating conditions (PLGA 1% (*w*/*v*), TWEEN85 1.5% (*v*/*v*), PVA_LMW_ 0.5% (*w*/*v*), and a 200 µM nozzle). As shown in Figure 3, the amount of loaded curcumin had a slight influence on particle size, with nanoparticles loaded with 10% CCM being smaller than those loaded with 20% CCM. Particle size and other physicochemical properties of nanoparticles are reported in Table 2. Nanoparticles loaded with curcumin exhibited high encapsulation efficiency, ranging between 66% and 76%, depending on the curcumin/PLGA ratio; the higher the ratio, the higher the EE.

Over the 4-week storage period set to verify the NPs stability, the curcumin content in the PLGA-NP9-CCM-10 formulation decreased from 3.76% to 2.54%, even when particles were stored at 4 °C, whereas the curcumin content in the PLGA-NP9-CCM-20 formulation remained stable under the same conditions. This reduction is likely attributable to the different distribution of curcumin on the nanoparticles surface. In fact, the lower surface charge observed in PLGA-NP9-CCM-20 suggests enhanced interaction between curcumin and PLGA carboxylate groups, which improved nanoparticle stability over time [47,48,52]. Blank nanoparticles exhibited a negative zeta potential due to the presence of carboxylic groups from PLGA. The loading of curcumin reduced the surface charge, indicating partial localization of curcumin on the nanoparticle surface. Notably, differences in zeta potential were observed with varying curcumin concentrations, with the formulation containing 20% curcumin showing increased interaction within the PLGA.

In order to verify the presence of any interaction between the PLGA and curcumin during the formation of the particles, FT-IR studies on curcumin, PLGA blank particles, and CCM-loaded nanoparticles were conducted. The FT-IR spectra of curcumin (Figure 4a) exhibit a peak at 3503 and 3085 cm^−1^ related to its OH group; the bands at 1625 and 1584 are attributed to the stretching vibration of C=O bonds and C=C aromatic rings, respectively. The bands at 1361 cm^−1^, 1235 cm^−1^, and 959 cm^−1^ correspond to the bending of the hydroxyl groups of the two phenolic and enolic groups, respectively, whereas the bands at 774 cm^−1^ and 1427 cm^−1^ are related to the olefinic in-plane bending vibrations of the heptadiene chain of curcumin [53]. Blank PLGA nanoparticles spectrum (Figure 4b) exhibits aliphatic C-H bonds stretching vibration at 2999–2859 cm^−1^, carbonyl C=O stretching vibrations at 1751 cm^−1^, and C-O stretching vibrations at 1087–1179 cm^−1^ [54], while the peak at 1448 cm^−1^ is related to the C-H stretching in a methyl group.

In the spectra of curcumin-loaded PLGA nanoparticles (Figure 4c–d), the C=O peak and the aromatic C=C are slightly shifted at higher wavenumber, 1630 cm^−1^ for the C=O, probably due to the formation of an intramolecular hydrogen bond, and at 1603 cm^−1^ related to π-π stacking or changes in the environment of curcumin’s aromatic rings. Moreover, the signal related to the C-H bond stretching vibrations at 2999–2859 cm^−1^ is slightly shifted at low wavenumber in the particles loaded with different amounts of curcumin due to hydrophobic interactions between the aromatic rings of curcumin and the non-polar aliphatic regions of PLGA. Such an effect is more pronounced in PLGA-NP9-CCM-20, probably due to the presence of a more extensive interaction between curcumin and the surface of the nanoparticles, increasing the intensity of the absorption bands in that region.

In vitro release studies were conducted using dialysis membrane (3.5 kDa molecular weight cut off) in a mixture composed of PBS (pH = 7.4 (70%)) and MeOH (30%) with 0.1% of Tween 80 to assess the release behavior of the encapsulated drug. Figure 5 shows representative release curves of curcumin from PLGA nanoparticles, produced by prilling in tandem with solvent evaporation with optimized operating conditions, loaded with different amounts of drug. Curcumin in vitro release from nanoparticles exhibited a burst within 6 h, with approximately 90% for PLGA-NP9-CCM-10 and 30% for PLGA-NP9-CCM-20. A two-phase release profile could be recognized for PLGA-NP9-CCM-20, with a rapid initial release (about 30%) likely due to stronger surface interactions, as indicated by FTIR analysis, and a sustained release attributed to PLGA degradation, leading to complete curcumin release within 7 days. Data fitting performed using Peppas–Korsmeyer’s equation [55] produced an r^2^_adj_ > 0.90 and a coefficient *n* of 0.71, indicating a complex non-Fickian transport mechanism probably involving diffusion and particle degradation.

## 4. Conclusions

This study successfully demonstrated the feasibility of using the prilling technique combined with solvent evaporation to produce submicrometric and nanometric PLGA particles with optimized properties. The optimization of solvent selection, surfactant concentration, and nozzle diameter proved critical in achieving stable and uniform particles with narrow size distributions (ranging from 316 nm to 462 nm). The selection of solvent and surfactant concentration emerged as critical factors significantly affecting particle size and stability. Due to its lower interfacial tension, ethyl acetate was identified as the optimal solvent, yielding more stable colloidal suspensions than dichloromethane.

Curcumin-loaded nanoparticles exhibited high encapsulation efficiencies (66–76%) and customizable release profiles, depending on drug loading. In vitro release studies showed distinct release profiles depending on drug loading. Formulations loaded with a lower amount of curcumin released 90% of their cargo within 6 h, while those loaded with a higher amount demonstrated a sustained release over 7 days, attributed to polymer degradation and stronger drug-polymer interactions, as confirmed by FTIR analysis.

These results suggest that prilling offers significant potential for developing advanced drug delivery systems, particularly for biomedical applications like wound healing and anti-inflammatory therapies. Future work could explore extending this methodology to other therapeutic agents and scaling up production for clinical applications.

## Figures and Tables

**Figure 1 pharmaceutics-17-00129-f001:**
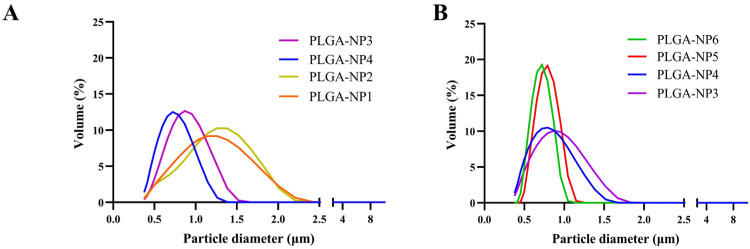
Size distribution of PLGA nanoparticles produced by prilling in tandem with solvent evaporation. (**A**) The effect of DCM and EtOAc in combination with PVA with different molecular weights (LMW and MMW). (**B**) The effect of different PVA (PVA_LMw_ and PVA_MMw_) at different concentrations using EtOAc as a solvent.

**Figure 2 pharmaceutics-17-00129-f002:**
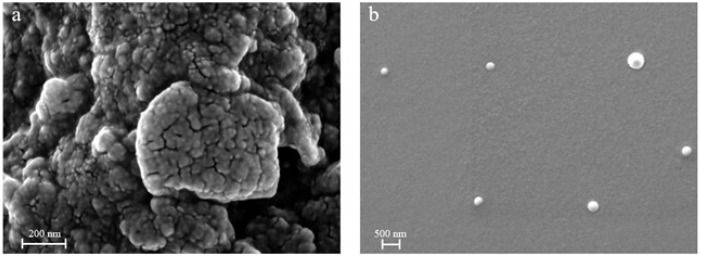
SEM microphotographs of PLGA nanoparticles produced by prilling with two different solvents for the feed: (**a**) DCM-based feed and (**b**) EtOAc-based feed.

**Figure 3 pharmaceutics-17-00129-f003:**
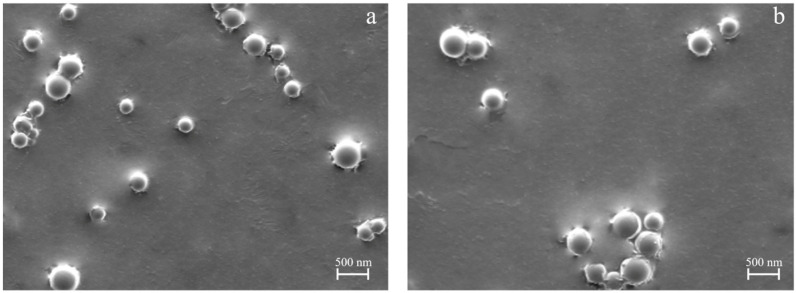
SEM microphotographs of PLGA nanoparticles produced by prilling with optimized conditions and loaded with different amounts of curcumin: (**a**) 10% (*w*/*w*) and (**b**) 20% (*w*/*w*) of the polymer.

**Figure 4 pharmaceutics-17-00129-f004:**
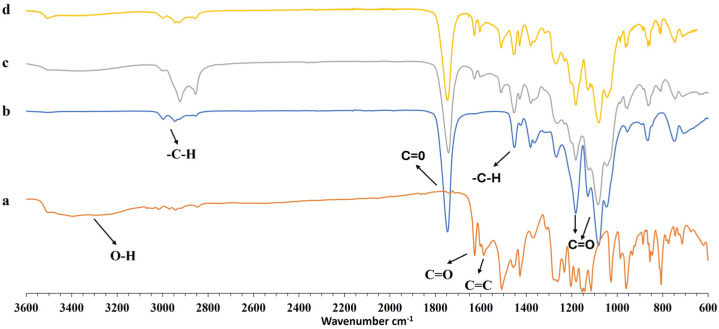
FTIR spectra of (**a**) curcumin, (**b**) blank PLGA nanoparticles, (**c**) PLGA-NP9-CCM-10, (**d**) PLGA-NP9-CCM-20.

**Figure 5 pharmaceutics-17-00129-f005:**
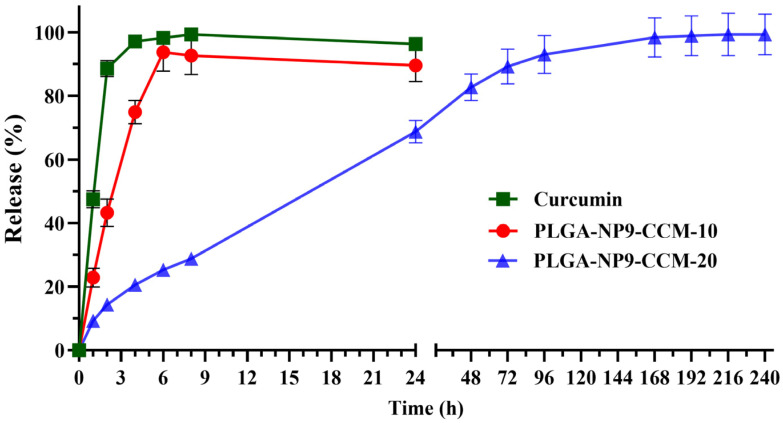
In vitro release of curcumin loaded into PLGA nanoparticles (PLGA-NP9-CCM-10 and PLGA-NP9-CCM-20) and curcumin alone (CCM) (mean ± SD, *n* = 3).

**Table 1 pharmaceutics-17-00129-t001:** Nanoparticles composition and prilling operative conditions.

Formulation Code	Solvent	Flow Rate (mL/min)	PVA_MMW_ (*w*/*v* %)	PVA_LMW_ (*w*/*v* %)	TWEEN85 (*v*/*v* %)	Nozzle (µM)	Frequency Hz
PLGA-NP1	DCM	3.0	1	-	3.0	200	270
PLGA-NP2	DCM	3.0	-	1.0	3.0	200	270
PLGA-NP3	EtOAc	8.0	1	-	3.0	200	270
PLGA-NP4	EtOAc	8.0	-	1.0	3.0	200	270
PLGA-NP5	EtOAc	8.0	0.5	-	3.0	200	270
PLGA-NP6	EtOAc	8.0	-	0.5	3.0	200	270
PLGA-NP7	EtOAc	8.0	-	0.5	3.5	200	270
PLGA-NP8	EtOAc	8.0	-	0.5	2.5	200	270
PLGA-NP9	EtOAc	8.0	-	0.5	1.5	200	270
PLGA-NP10	EtOAc	8.0	-	0.5	0.5	200	270
PLGA-NP11	EtOAc	2.0	-	0.5	1.5	120	300
PLGA-NP12	EtOAc	2.5	-	0.5	1.5	80	350

**Table 2 pharmaceutics-17-00129-t002:** Physicochemical properties of blank PLGA(PLGA-NP9) and curcumin-loaded PLGA nanoparticles (PLGA-NP9-CCM-10 and PLGA-NP9-CCM-20) obtained by prilling in tandem with solvent evaporation (mean ± SD, *n* = 3).

Formulation Code	Size (nm)	PDI	Zeta Potential (mV)	Drug Content (%)	EE (%)
At time 0					
PLGA-NP9	316 ± 90	0.19 ± 0.03	−34.11 ± 0.57	-	-
PLGA-NP9-CCM-10	391 ± 77	0.33 ± 0.01	−31.18 ± 1.10	3.76 ± 0.05	76.10 ± 0.20
PLGA-NP9-CCM-20	462 ± 22	0.57 ± 0.07	−28.00 ± 0.44	8.72 ± 0.18	66.44 ± 1.41
After 28 Days					
PLGA-NP9	486 ± 123	0.27 ± 0.09	−33.42 ± 0.79	-	-
PLGA-NP9-CCM-10	337 ± 51	0.17 ± 0.09	−36.09 ± 0.92	2.54 ± 0.02	51.42 ± 0.24
PLGA-NP9-CCM-20	493 ± 89	0.28 ± 0.04	−22.50 ± 1.90	unchanged	unchanged

## Data Availability

The data presented in this study are available on request from the corresponding author.

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
