# Peer review of "Prilling as an Effective Tool for Manufacturing Submicrometric and Nanometric PLGA Particles for Controlled Drug Delivery to Wounds: Stability and Curcumin Release"

_pharmaceutics, 2025, doi:10.3390/pharmaceutics17010129_

Round 1

Reviewer 1 Report

Comments and Suggestions for Authors

The manuscript needs substantial corrections as outlined below.

Authors should incorporate literature reports on the production of PLGA nanoparticles using other techniques to give a comparative information for better understanding of the study. Also please cite other studies where prilling is used to make PLGA nanoparticles

A paragraph discussing the prior studies related to PLGA encapsulated curcumin should be included in the discussion.

SEM of all the samples including curcumin loaded NPs should be provided in the manuscript.

In addition to this, the manuscript requires improvement in language and content clarity. Few specific corrections include:

Line 51: Correct the spelling of "properties."

Line 139: Clarify the abbreviation "CCM."

Line 165: Revise the line: “between polymers and between polymers and drugs were…”

Lines 219, 224: Correct the abbreviation for ethyl acetate.

Table 1 caption shpuld be revised indicating solvent, flow rate and concentration of PVA, Nozzle and frequency in each row.

The caption of figure 3 needs to be revised with more clarity.

Comments on the Quality of English Language

Should be improved

Author Response

The authors would like to thank the reviewer for the insightful comments on the manuscript that have been useful to improve and present the work. The specific comments raised by the reviewer have been punctually addressed and the authors’ comments are hereunder reported.

Comments 1:The authors should incorporate literature reports on the production of PLGA nanoparticles using other techniques to give a comparative information for better understanding of the study. Also please cite other studies where prilling is used to make PLGA nanoparticles.

Response 1 : Both introduction and Results and Discussion sessions have been edited in order to include more comprehensive information about PLGA nanoparticles production and the use of prilling to produce beads and microparticles, including literature references (Cun et al, European Journal of Pharmaceutics and Biopharmaceutics 2011, 77, 26-35; Albisa et al, Pharmaceutical Research 2017, 34, 1296-1308; Andreana et al., International Journal of Molecular Sciences 2023, 24). It was not possible to include references in the use of prilling technique to produce PLGA nanoparticles as this work represents the first attempt to do so, in our knowledge.

Comments 2: A paragraph discussing the prior studies related to PLGA encapsulated curcumin should be included in the discussion.

Response 2 : The Results and Discussion session has been edited in order to include brief considerations and literature references related to PLGA particles loaded with curcumin (Chereddy et al., Journal of Controlled Release 2013, 171, 208-215; Verderio et al., Biomacromolecules 2013, 14, 672-682).

Comments 3: SEM of all the samples including curcumin loaded NPs should be provided in the manuscript.

Response 3 : The Results and Discussion session has been improved with the addition of SEM microphotographs of curcumin loaded PLGA nanoparticles as a new Figure 3. Moreover, the paragraph concerning the preparation of curcumin loaded NPs has been improved with the following text “In order to investigate the impact of different curcumin concentrations on the physico-chemical properties of the nanoparticles, two specific curcumin amounts were tested. Curcumin (CCM) at 10% and 20% w/w was incorporated into PLGA nanoparticles by directly adding CCM to the PLGA feed solution, followed by prilling under optimized operating conditions (PLGA 1% (w/v), TWEEN85 1.5% (v/v), PVALMW 0.5% (w/v), and a 200 µm nozzle). As shown in Figure 3, the amount of loaded curcumin had a slight in-fluence on particle size, with nanoparticles loaded with 10% CCM being smaller than those loaded with 20% CCM. Particle size and other physicochemical proprieties of nanoparticles are reported in Table 2."

Comments 4: In addition to this, the manuscript requires improvement in language and content clarity. Few specific corrections include:

  • Line 51: Correct the spelling of "properties."
  • Line 139: Clarify the abbreviation "CCM."
  • Line 165: Revise the line: “between polymers and between polymers and drugs were…”
  • Lines 219, 224: Correct the abbreviation for ethyl acetate.

Response 4 : The Authors thank once again the reviewer for the suggestions. All the corrections suggested by the reviewer have been made and the whole manuscript has been improved in terms of language and editing.

Comments 5: Table 1 caption should be revised indicating solvent, flow rate and concentration of PVA, Nozzle and frequency in each row.

Response 5 : Table1 has been re-edited according to the suggestions received.

Comments 6: The caption of figure 3 needs to be revised with more clarity.

Response 6 : Caption of former Figure 3, now Figure 4 has been edited in order to enhance its clarity…”Figure 4. FTIR spectra of curcumin raw material (a) and blank PLGA nanoparticles (d) in comparison to PLGA nanoparticles loaded with different amount of curcumin PLGA- NP9-CCM-20 (b) and PLGA-NP9-CCM-10 (c).”

Reviewer 2 Report

Comments and Suggestions for Authors

Dear Editor and Authors,

I read the paper carefully, and it is interesting. However, before accepting the publication, some changes must be discussed. The comments are presented below.

1.     Please present in the Introduction part the study's novelty compared to the recent literature.

2.     Tween 85 at different concentrations (0.5; 1.5; 2.5; 3; 3.5 % v/v) were added to the solution to avoid the aggregation of particles. I don’t understand why adding the emulsifier is necessary if the final is about a solution, not an emulsion.

3.     Please add the PVA solvent, probably water, but mention in the paper is necessary.

4.     Also, it is evident that the aggregates are formed in DCM, taking into account that DCM is not compatible with water.

5.      Please explain why the curcumin-loaded nanoparticles have negative Zeta Potential.

6.     Please add more SEM images with the capsules.

Author Response

The authors would like to thank the reviewer for the insightful comments on the manuscript that have been useful to improve and present the work. The specific comments raised by the reviewer have been punctually addressed and the authors’ comments are hereunder reported.

Comments 1: Please present in the Introduction part the study's novelty compared to the recent literature.

Response 1 : Introduction and Conclusions sessions have been edited in order to clearly underline the novelty of the work and the use of prilling to produce PLGA nanoparticles. More references related to PLGA nanoparticles production and the use of prilling to produce beads and microparticles, including literature references have been added  (Cun et al, European Journal of Pharmaceutics and Biopharmaceutics 2011, 77, 26-35; Albisa et al, Pharmaceutical Research 2017, 34, 1296-1308; Andreana et al., International Journal of Molecular Sciences 2023, 24; Séquier et al. European Journal of Pharmaceutics and Biopharmaceutics 2014, 87, 530-540; Beltrame et al. Food Chemistry 2024, 460, 140694.)

Comments 2: Tween 85 at different concentrations (0.5; 1.5; 2.5; 3; 3.5 % v/v) were added to the solution to avoid the aggregation of particles. I don’t understand why adding the emulsifier is necessary if the final is about a solution, not an emulsion.

Response 2: The Authors want to apologies for the insufficient clarity reported in the manuscript. The Results and Discussion and References sections have been edited in order to better clarify the use of Tween 85 with the following text“TWEEN 85 is a non-ionic surfactant, able to interact with hydrophobic and hydrophilic components, commonly used to stabilize nanosuspensions preventing the aggregation by forming a steric barrier around the nanoparticles [50].

  1. Sun, W.; Xie, C.; Wang, H.; Hu, Y. Specific role of polysorbate 80 coating on the targeting of nanoparticles to the brain. Biomaterials 2004, 25, 3065-3071, doi:https://doi.org/10.1016/j.biomaterials.2003.09.087.

Comments 3: Please add the PVA solvent, probably water, but mention in the paper is necessary.

Response 3: The paragraph “Nano and submicrometric particles preparation and Preparation of feed solution containing curcumin” in the Materials and Methods section has been improved specifying the aqueous nature of the PVA solution. The following text has been added/modified “A laminar jet of polymer solution is broken into droplets by vibration, falling down into the collecting solution composed by an aqueous solution of poly-vinyl alcohol (PVA) in different concentrations"….and  "….The drops falling into collecting solution composed by an aqueous solution of PVALMW 0.5% (w/v) solidify by solvent evaporation to form PLGA nanoparticles”

Comments 4: Also, it is evident that the aggregates are formed in DCM, taking into account that DCM is not compatible with water.

Response 4: Authors thank once again the reviewer for the comments. The Results and Discussion section have been improved including discussion about the DCM effect on particles due to its properties. The following text has been added “After a first series of preliminary experiments used to set some of the operational parameters, the impact of the organic solvent on PLGA mean particle size was investigated. DCM (immiscible in water), one of the most commonly used solvent to prepare PLGA nanoparticles by solvent evaporation method [46] and EtOAc (slightly soluble in water) were used”.

  1. Alkholief, M.; Kalam, M.A.; Anwer, M.K.; Alshamsan, A. Effect of Solvents, Stabilizers and the Concentration of Stabilizers on the Physical Properties of Poly(d,l-lactide-co-glycolide) Nanoparticles: Encapsulation, In Vitro Release of Indomethacin and Cytotoxicity against HepG2-Cell. Pharmaceutics 2022, 14, doi:10.3390/pharmaceutics14040870.

Comments 5: Please explain why the curcumin-loaded nanoparticles have negative Zeta Potential

Response 5: The Results and Discussion session has been improved highlighting the effect of curcumin loading in the nanoparticles and how it is correlated to some of the nanoparticle properties “Blank nanoparticles exhibited a negative zeta potential due to the presence of carboxylic groups from PLGA. The loading of curcumin reduced the surface charge, indicating partial localization of curcumin on the nanoparticle surface. Notably, differences in zeta potential were observed with varying curcumin concentrations, with the formulation containing 20% curcumin showing increased interaction within the PLGA”.

Comments 6: Please add more SEM images with the capsules

Response 6: As suggested additional SEM microphotographs of curcumin loaded PLGA nanoparticles have been added in a new Figure 3. Moreover, the paragraph concerning the preparation of curcumin loaded NPs has been improved with the following text “… As shown in Figure 3, the amount of loaded curcumin had a slight in-fluence on particle size, with nanoparticles loaded with 10% CCM being smaller than those loaded with 20% CCM. Particle size and other physicochemical proprieties of nanoparticles are reported in Table 2."

Round 2

Reviewer 1 Report

Comments and Suggestions for Authors

The Manuscript needs few minor corrections:

Please correct the caption of figure 4 "figure 4. FTIR spectra of curcumin raw material (a) and blank PLGA nanoparticles (d) in comparison to PLGA nanoparticles loaded with different amount of curcumin PLGA- NP9-CCM-20 (b) and PLGA-NP9-CCM-10 (c). " it would be better if authosr could write teh caption as (a) Cucrumin (b) PLGA ananoparticles  (c) ......... In the current text its confusing to identify the corresponding spectra. Also, authors are encouraged to follow similar pattern in writing captions for all the figures.

Manuscript needs grammatical corrections at few places. for instance:

Line 336 "shifted at low wavenumber in both PLGA-NP9-CCM particles due to hydrophobic interactions..." probably word "both" is not required here.

Comments on the Quality of English Language

No.

Author Response

The authors would like to thank again the reviewer for the insightful comments on the manuscript that have been useful to improve and present the work. The specific comments raised by the reviewer have been punctually addressed and the authors’ comments are hereunder reported.

Comment 1: Please correct the caption of figure 4 "figure 4. FTIR spectra of curcumin raw material (a) and blank PLGA nanoparticles (d) in comparison to PLGA nanoparticles loaded with different amount of curcumin PLGA- NP9-CCM-20 (b) and PLGA-NP9-CCM-10 (c). " it would be better if authosr could write teh caption as (a) Cucrumin (b) PLGA ananoparticles  (c) ......... In the current text its confusing to identify the corresponding spectra. Also, authors are encouraged to follow similar pattern in writing captions for all the figures.

Response 1 : Caption of Figure 4 has been edited in order to enhance its clarity.Figure 4. FTIR spectra of (a) curcumin, (b) PLGA- NP9-CCM-20, (c) PLGA-NP9-CCM-10 and (d) blank PLGA nanoparticles.” Moreover, to maintain a consistent writing style across all captions, Figures 2 and 3 were also modified. “Figure 2. SEM microphotographs of PLGA nanoparticles produced by prilling with two different solvents for the feed: (a) DCM based feed and (b) EtOAc based feed.” Figure 3. SEM microphotographs of PLGA nanoparticles produced by prilling with optimised conditions and loaded with different amount of curcumin: (a) 10% (w/w) and (b) 20% (w/w) of the polymer”.

Comment 2:  Line 336 "shifted at low wavenumber in both PLGA-NP9-CCM particles due to hydrophobic interactions..." probably word "both" is not required here.

Response 2  : The Authors thank once again the reviewer for the suggestions. The use of the word "both" was referred to the signal shift present in the two curcumin loaded nanoparticle formulations. Hojhever, to better clarify the explanation of FTIR dara we reformulated the sentence in: “ Moreover, the signal related to the C-H bond stretching vibrations at 2999–2859 cm¹ are slightly shifted at low wavenumber in the particles loaded with different amount of curcumin due to hydrophobic interactions between the aromatic rings of curcumin and the non-polar aliphatic regions of PLGA.”

Reviewer 2 Report

Comments and Suggestions for Authors

Accept

Author Response

The authors would like to thank the reviewer for the comments on the manuscript that have been useful to improve the presented work.